# Feasibility and Accuracy of the Automated Software for Dynamic Quantification of Left Ventricular and Atrial Volumes and Function in a Large Unselected Population

**DOI:** 10.3390/jcm10215030

**Published:** 2021-10-28

**Authors:** Gianpiero Italiano, Gloria Tamborini, Laura Fusini, Valentina Mantegazza, Marco Doldi, Fabrizio Celeste, Paola Gripari, Manuela Muratori, Roberto M. Lang, Mauro Pepi

**Affiliations:** 1Centro Cardiologico Monzino IRCCS, 20138 Milan, Italy; gloria.tamborini@ccfm.it (G.T.); laura.fusini@ccfm.it (L.F.); Valentina.Mantegazza@ccfm.it (V.M.); doldi.mrc@gmail.com (M.D.); fabrizio.celeste@ccfm.it (F.C.); Paola.Gripari@ccfm.it (P.G.); Manuela.Muratori@ccfm.it (M.M.); mauro.pepi@ccfm.it (M.P.); 2Department of Medicine, University of Chicago Medical Center, Chicago, IL 60637, USA; rlang@medicine.bsd.uchicago.edu

**Keywords:** 3D echocardiography, cardiac chamber quantification, left ventricular function, machine learning

## Abstract

We aimed to evaluate the feasibility and accuracy of machine learning-based automated dynamic quantification of left ventricular (LV) and left atrial (LA) volumes in an unselected population. We enrolled 600 unselected patients (12% in atrial fibrillation) clinically referred for transthoracic echocardiography (2DTTE), who also underwent 3D echocardiography (3DE) imaging. LV ejection fraction (EF), LV, and LA volumes were obtained from 2D images; 3D images were analyzed using dynamic heart model (DHM) software (Philips) resulting in LV and LA volume–time curves. A subgroup of 140 patients also underwent cardiac magnetic resonance (CMR) imaging. Average time of analysis, feasibility, and image quality were recorded, and results were compared between 2DTTE, DHM, and CMR. The use of DHM was feasible in 522/600 cases (87%). When feasible, the boundary position was considered accurate in 335/522 patients (64%), while major (n = 38) or minor (n = 149) border corrections were needed. The overall time required for DHM datasets was approximately 40 seconds. As expected, DHM LV volumes were larger than 2D ones (end-diastolic volume: 173 ± 64 vs. 142 ± 58 mL, respectively), while no differences were found for LV EF and LA volumes (EF: 55% ± 12 vs. 56% ± 14; LA volume 89 ± 36 vs. 89 ± 38 mL, respectively). The comparison between DHM and CMR values showed a high correlation for LV volumes (r = 0.70 and r = 0.82, *p* < 0.001 for end-diastolic and end-systolic volume, respectively) and an excellent correlation for EF (r = 0.82, *p* < 0.001) and LA volumes. The DHM software is feasible, accurate, and quick in a large series of unselected patients, including those with suboptimal 2D images or in atrial fibrillation.

## 1. Introduction

An accurate and reproducible quantitative assessment of the left heart chamber’s size and function is pivotal for the diagnosis, treatment, and prognosis of heart diseases. In this regard, the most important contribution of 3D transthoracic echocardiography (3DE) is left ventricular (LV) and left atrial (LA) quantification.

Recent development of artificial intelligence approaches, such as machine learning (ML) algorithms, has resulted in a 3D analysis technique that automatically detects LV and LA boundaries and allows fast, accurate, automated measurements of chamber volumes and function. While several studies have validated the “static” heart model (HM) [1], more recently, a dynamic HM (DHM), which generates end-diastolic and end-systolic LV and LA volumes [2] as well as volume–time curves of these chambers, has been developed [3]. However, this recent methodology has been reported only in small groups of selected patients who have good-quality images. To the best of our knowledge, this is the first study performing this 3D analysis in a large, unselected population.

The aims of our study were: (a) to evaluate the feasibility of the new DHM software in a large group of unselected patients with a wide range of acoustic windows, irrespective of heart rhythm; (b) to evaluate the relationship between image quality and accuracy of the detection of LV and LA boundaries; (c) to compare 2D echocardiography and DHM-derived LV and LA volumes and function and in a subgroup of cases to cardiac magnetic resonance (CMR) reference values.

## 2. Materials and Methods

A total of 600 patients (age 63 ± 14 years; 431 men and 169 women) referred for TTE were prospectively enrolled in this study from January 2019 to February 2020. All subjects underwent a complete 2D and 3D TTE. Moreover, 140 patients also underwent CMR for diagnostic reasons. The overall population included 40 patients with normal hearts, 312 patients with valve disease, 58 with coronary artery disease, 113 with dilated cardiomyopathy, and 77 patients with other miscellaneous pathologies (including 6 with hypertrophic cardiomyopathy). The study protocol complied with the ethical guidelines of the 1975 Declaration of Helsinki and was approved by our institutional review board. Informed consent was obtained from each patient.

### 2.1. Two-Dimensional Echocardiography

All echocardiographic examinations were performed using a Philips EPIQ echocardiographic system (Philips Healthcare, Andover, MA) equipped with an X5-1 transducer. LV end-diastolic volume (EDV), LV end-systolic volume (ESV), LV ejection fraction (EF) and LA volumes were measured from the 4- and 2-chamber views using the biplane Simpson’s method according to international guidelines.

#### Automated Three-Dimensional Echocardiography

Following 2D echocardiographic (2DE) imaging, 3D full-volume acquisitions were obtained with the same probe from the 4-chamber apical view with the LV and LA properly centered along the major axis, as previously described [3,4,5]. Before each acquisition, image settings were optimized by modifying the gain, compress, and time gain compensation controls to achieve the highest possible frame rate.

Briefly, 3D datasets were acquired in a single beat during a breath hold lasting a few seconds, ensuring optimal temporal and spatial resolution. The volumetric datasets were immediately evaluated on-board using the DHM software (Philips Healthcare), which automatically identifies LV endo- and epi-cardial borders at end-diastole and LA borders at end-systole, allowing prompt quantification of the volumes of these chambers (Figure 1). In our study, 3DE images were analyzed using the default settings of the boundary detection sliders (end-diastolic default position = 60/60; end-systolic default position = 30/30). We categorized DHM data into three main categories depending on image and waveform quality: good, suboptimal, and poor. Qualitative criteria of acquisition/display of volumes and volume curves were: (a) optimal/suboptimal definition of atrial and ventricular shape including the entire LV and LA cavities without artefacts and optimal/suboptimal recognition of the mitral annulus (with simultaneous and opposite volume changes in LA vs. LV throughout the cardiac cycle); (b) LV volume–time curves depicting systolic contraction followed by biphasic filling with a period of diastasis separating the active, rapid LV filling phase from the passive LV filling phase as a result of atrial contraction; (c) LA volume–time curves depicting atrial filling followed by biphasic emptying with a quiescent period of diastasis. Good and suboptimal data were analyzed, while poor images were considered unfeasible and discarded. Major and minor adjustments of LV endocardial and epicardial borders were then performed manually as needed to better refine endocardial border tracking. The LV and LA volume–time curves obtained using the automated software were recorded, and the different phases of the cardiac cycle were identified using maximum and minimum volumes and their derivative curves [3]. All investigators who performed echocardiographic analyses were blinded to the CMR results.

### 2.2. CMR Protocol

CMR imaging was performed using a 1.5-T scanner (Discovery MR450; GE Healthcare, Milwaukee, WI, USA) within 7 days from the echocardiographic examination to ensure that identical hemodynamic conditions were not made during changes in therapeutic regimens. All studies were carried out using dedicated cardiac software, phased-array surface receiver coils, and electrocardiogram triggering. Breath-hold steady-state free-precession cine imaging was performed in long- and short-axis orientations. A stack of short-axis slices encompassing right and left chambers from base to apex was used for ventricular and atrial volumes and systolic function assessment. CMR datasets were transferred to a dedicated workstation and analyzed with a cardiac software (Report Card version 4.0, GE Healthcare, Chicago, IL, USA). LVEDV, LVESV, LVEF, and LA volume were evaluated on cine images according to the recommendations of the Society of Cardiovascular Magnetic Resonance [6].

### 2.3. Reproducibility

To assess the reproducibility of DHM measurements, intraobserver and interobserver variability were evaluated in a subset of 15 randomly selected patients. Physicians were blinded to the original LA, LV, and EF results.

To derive interobserver variability, one physician was asked to recalculate DHM acquisition for 15 patient datasets after 4 weeks. Two investigators (G.I and V.M.) were asked to perform these measurements twice to assess intraobserver variability. The same loops were re-analyzed 2 weeks later by the two investigators who were blinded to all previous measurements. These investigators repeated the analysis with and without contour adjustments.

### 2.4. Statistical Analysis

Results are presented as mean ± standard deviation (SD). Volumes and LVEF data obtained with the DHM were compared with 2D and CMR values by using linear regression analysis with Pearson’s correlation coefficients. Moreover, Bland–Altman analysis was applied to evaluate the inter-technique agreement by calculating the bias (mean difference) and the 95% limits of agreement (defined as 1.96 SD around the mean difference). To assess the association between the categorical variables, a chi-square test was applied. Differences were considered significant with *p*-values < 0.05. Statistical analysis was performed using SPSS 25 (SPSS Inc., Chicago, IL, USA).

## 3. Results

Baseline characteristics of the study patients are summarized in Table 1. DHM analysis of the LV was feasible in 522 patients (87%), while 78 3D datasets were discarded due to incorrect automatic identification of LV/LA borders. Feasibility was high in all categories and pathologies but slightly lower in patients with atrial fibrillation (81%) and cases with coronary artery disease (81%) compared to normal subjects (97%) (Table 2). LV quantification using the automated software required major adjustment of LV borders in 38/522 patients (7%), whereas minor adjustments were required in 149/522 patients (29%). We classified DHM datasets depending on the acquisition quality and volume waveform quality as good (*n* = 327), suboptimal (*n* = 195), and poor (*n* = 78). The average time needed for DHM acquisition and analysis was 41 ± 12 seconds using automated software with manual corrections as needed. Seventy-three patients showed irregular rhythms at examination time, secondary to atrial fibrillation (Table 2). Need for adjustments was higher in patients with dilated cardiomyopathy vs. normal subjects (12%, *p* = 0.13) due to incorrect border identification, especially in extremely dilated cavities. 

Table 3 summarizes the results of the comparisons between 2D TTE (a) and CMR (b) measurements of LVEDV, LVESV, LVEF, and LA volumes with the corresponding DHM values. Despite larger LV volumes measured by DHM compared to 2D TTE, we noticed a significant correlation between 2D TTE and DHM values (LVEDV: *r* = 0.94, *p* < 0.001, LVESV: *r* = 0.96, *p* < 0.001) showing excellent agreement. The automated algorithm resulted in minimal differences for LVEF, reflected by a minimum negative bias of 1.2% (Figure 2). No significant differences were found in LA volume quantification between 2DTTE and DHM (*p* = 0.023) and DHM and MRI.

Figure 3 shows the comparison between DHM and CMR values. The results of analysis revealed a moderate correlation (LVEDV: *r* = 0.70, *p* < 0.001; LVESV: *r* = 0.82, *p*<0.001) with a reasonable bias for LV volumes due to DHM underestimation with a good correlation for LVEF (*r* = 0.82, *p* <0.001) with a slight overestimation by the automated algorithm with a narrow bias of 2.0%. Correlations between DHM and CMR for LA volumes were high with minimal bias.

Figure 4 shows examples of automated volume–time waveforms in different subgroups of patients and their typical pathological curves. 

## 4. Discussion

This is the first study to test the feasibility and accuracy of this automated software in a large, unselected population. The main findings of this study are as follows: (1) the dynamic heart model automated measurement of LV and LA volumes and function is highly feasible in a large, unselected population, and analysis duration is very short; (2) independent from pathologies and heart rhythm, the quality of DHM images is good, with only a minority of cases requiring major manual adjustment of boundaries; (3) DHM LV volume values had good agreement with biplane 2D measurements despite an overestimation of both diastolic and systolic LV volumes, and the same is true in terms of the correlation between DHM values vs. CMR with a DHM underestimation of LV volumes; (4) LVEF correlation among the three techniques was very high; (5) measurements of LA volumes by DHM showed excellent agreement with CMR volumes. Several studies validated the heart model-automated measurements of LV and LA volumes [1,7,8], and optimal settings for boundary detection have been defined [2]. Previously, this was limited to end-systolic and end-diastolic frames, with the dynamic volume changes that occur during the cardiac cycle being ignored. In a small, selected population, the feasibility and accuracy of the first dynamic 3D ML analysis software have recently been validated [1]. This dynamic analysis provides frame-by-frame measurements of left heart chamber volumes. Despite minor manual corrections of the automatically detected LV and LA borders, analysis was performed in 4/20 and 5/20 cases, respectively, and the time required for generation of time–volume curves of LV and LA from 3D was very short (60 ± 30 sec per patient) [3].

In this study, we found that DHM was highly feasible and rapid (about 40 seconds) in 600 unselected cases, and the quality of data was good in patients with a variety of pathologies and cardiac rhythms, such as atrial fibrillation requiring no or only minor boundary adjustments in the large majority (94%) of patients. DHM feasibility in the entire non-selected population was 87%, similar to previous studies with ML-based techniques [9,10,11]. Indeed, HM failed in about 10% of the patients, and this failure rate of the automated algorithm is comparable to the percentage of patients with substandard echocardiographic images. Interestingly, in normal subjects, feasibility was very high (97%), and this method may further reduce the examination time in “simple” cases. 

Manual editing of the automatically detected LV endocardial surfaces is possible when the operator judges that the automatically detected surface is incorrect. According to our experience, manual border corrections are required in patients with suboptimal acoustic windows and in patients with pathologies in which anatomical deformation made automated endocardial tracking more complicated, such as dilated cardiomyopathy and hypertrophic cardiomyopathy. In these cases, minimal modifications of automated cardiac boundaries provide accurate volumetric analysis compared to both the gold-standard CMR reference and the conventional 2D echocardiography. An important advantage of automated software is that operator training is not required. Small adjustments of cardiac boundaries require less expertise than tracing the boundaries manually in conventional echocardiography. Furthermore, our results show excellent reproducibility of this approach, as expected for an automated technique, and results in significant time saving compared to the conventional analysis.

Even though the accuracy of LV and LA functional curves was beyond the scope of our study, in the quality score, we included the plausibility of curves, and examples in our series clearly show that the systolic and diastolic phases of LV and LA curves are well defined. Specifically, in sinus rhythm and atrial fibrillation, patterns and profiles of LV and LA volume curves identify the presence or absence of atrial contraction [12]. Normal subjects, patients with dilated ventricles and systolic dysfunction, and patients with LV volume overload and normal systolic function (i.e., mitral regurgitation) had peculiar LV and LA curves consistent with these different conditions. In the future, an automated analysis of these curves may help to better recognize pathological changes and provide additional useful information in different conditions.

With regards to feasibility, quality, and the percentage of major or minor adjustments, in normal subjects, valvular heart disease and coronary artery disease showed similar good DHM quality and need for minor adjustments. On the contrary, in dilated cardiomyopathy and miscellaneous pathologies, manual minor or major adjustments were needed in approximatively 60% of the cases. This observation is in agreement with previous data showing that the accuracy of HM volumes was lower in some pathologies because of incorrect detection of LV endocardium. Specifically, in some cases, the automatic detection of LV and LA borders may lead to the incorrect identification of the atrioventricular junction (i.e., presence of a sigmoidal shape of the interventricular septum that was interpreted by the software as mitral annulus, or cases with normal LV and very large LA), or very dilated LV in which the apical and antero-lateral endocardium is not perfectly identified. Regarding patients with atrial fibrillation, the percentage of good images or minor adjustment was low (48%) despite high feasibility (81%).

Measurements of 2D biplane standard LV and LA volume versus 3D DHM showed good agreement as demonstrated in previous studies [2,7,13]. All these data are in agreement with previous studies and meta-analysis comparing 2D and 3D echocardiography in measuring LV volumes and EF [7,9,10,11,14,15,16]. The underestimation of LV volumes with 3DE versus CMR is still under debate, and the following aspects should be considered: (a) superior image quality of CMR, with better endocardial border definition, as well as the different tracing methods of 3DE and CMR; (b) in CMR, LV volumes are obtained with the endocardial border traced inside the trabeculations, whereas in 3DE, it is often traced where the trabeculations meet the LV cavity; (c) the inclusion of basal LV slices may also play a role in the larger LV volumes obtained with CMR [13,14].

LA size and function parameters are associated with adverse outcomes in multiple disease states. Recent data suggest that the phasic LA function also holds prognostic information [17,18,19,20,21]. Conventional 2D TTE is known to systematically underestimate LA volume compared with CMR [14,15,16]. When compared with CMR reference standards, 3DE provides more accurate and reproducible quantifications of LA volumes [22]. Our data clearly show that DHM allows accurate and rapid measurements of LA volume and function. Correlations between DHM volumes and CMR were excellent with minimal bias.

Despite the demonstrated benefits of 3DE over 2D TTE [4], widespread use of 3DE for heart chamber volume assessment has not yet become a clinical reality. This scenario is likely due to the time and training required to obtain accurate and reproducible 3DE volumetric measurements [23,24]. Previous studies clearly demonstrated that HM was feasible, very rapid, and accurate. Once the 3DE dataset is loaded, the software used in the present study does not require any input to automatically perform the measurements. Moreover, after launching the acquired 3DE dataset, the examiner immediately displays: LA and LV contours on four-, three-, and two-chamber cut-planes extracted from the 3DE datasets; moving casts of the LV and LA with the perception of the ideal recognition of the two cavities; and LV/LA reciprocal volumetric changes throughout the cardiac cycle. The different phases of the cardiac cycle were identified using maximum and minimum values of the volume curves and their time-derivative LV and LA curves. In cases where the operator felt that the automatically detected surface was incorrect, manual corrections were easily performed.

Tsang et al. [1] first described the heart model method based on LV and LA models built by using automated endocardial surface detection in conjunction with information from a large 3D echocardiographic database. Casts of the LV and LA were derived without geometrical assumptions. Importantly, to avoid problems related to foreshortening, LV- and LA-focused apical two- and four-chamber views were identified as those in which the long-axis dimension of the relevant chamber was maximized. This method is not only ideal for the LV but also for the LA since it is not easy to obtain the correct four- and two-chamber views in the standard 2D examinations; therefore, the biplane formula may be suboptimal. LA volume measurements by standard 2D examination should be performed in LA-focused views to avoid foreshortening and, thus, underestimating volumes. The new software automatically detects the LA boundaries in 3D space and measures the actual LA volume without any geometrical assumptions, and, thus, it performs more accurately than 2D methodology.

The use of automated analysis software significantly reduced the time required to obtain LVEF, LV, and LA volumes with excellent reproducibility. As shown in Figure 5, interobserver variability for 3D measurements was minimal, depending on manual adjustments needed in the case of inadequate border identification. These results are extremely important for future applications of automated analysis; its introduction in daily practice may avoid reader measurement variability, allowing better assessment of volume and function changes.

### Limitations

Machine learning software requires a high-quality 3D dataset, thus restraining its application in patients with inadequate TTE windows. Although the use of the DHM algorithm showed high feasibility of left chamber analysis, automated border detection was difficult in some of the datasets. With 3DE imaging, specifically in single-beat acquisitions, low-frame-rate datasets may miss the true end-systolic frame. Despite the attempt to maximize frame rates, it was not always possible to ensure that it was sufficiently high to avoid this pitfall.

## 5. Conclusions

Our findings confirm that incorporating DHM into daily practice is a promising way to overcome challenges associated with the previous 3D clinical workflow.

## Figures and Tables

**Figure 1 jcm-10-05030-f001:**
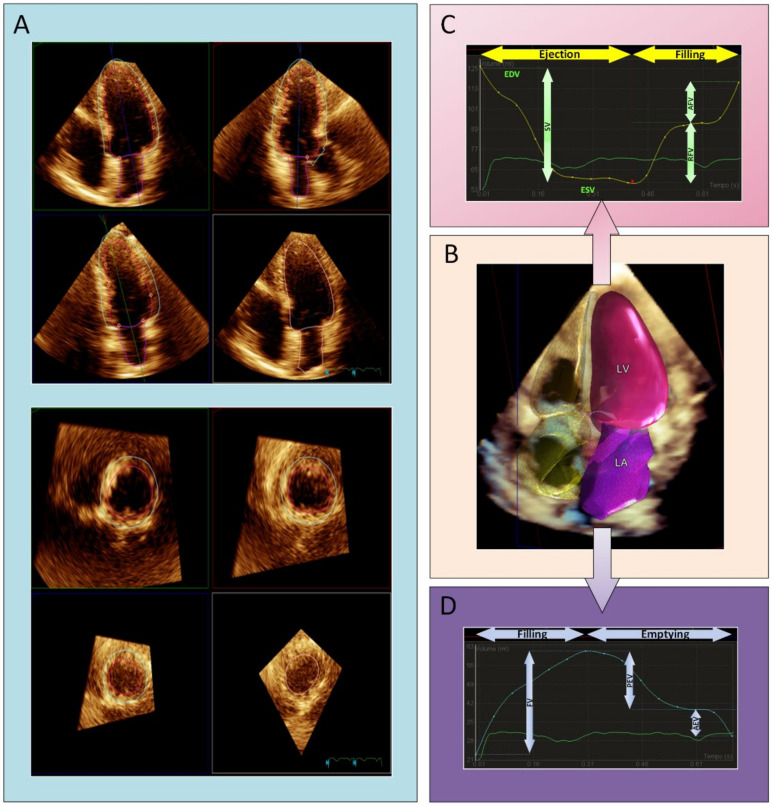
(**A**) DHM automated analysis of left ventricular and left atrial volumes in apical view. (**B**) 3D surface rendering of the LV (pink and semitransparent blue) and LA (purple) at end-diastole. (**C**) Analysis of LV volume–time curves, resulting in dynamic ejection and filling, end-diastolic (EDV) and end-systolic volumes (ESV), stroke volume (SV), rapid filling volume (RFV), and atrial filling volume (AFV). (**D**) Analysis of LA volume–time curves, resulting in dynamic filling and emptying: filling volume (FV), passive emptying volume (PEV), and active emptying volume (AEV). LA = left atrium; LV = left ventricle.

**Figure 2 jcm-10-05030-f002:**
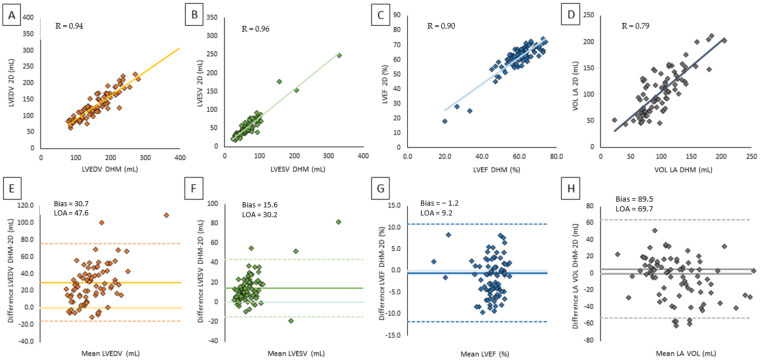
Correlation and agreement between 2D TTE and DHM on left chamber volume and LV function: (**A**) scatter plot of LVEDV by 2DTTE versus DHM, (**B**) scatter plot of LVESV by 2D TTE versus DHM, (**C**) scatter plot of LA volume by 2D TTE versus DHM, (**D**) scatter plot of LVEF by 2DTTE versus DHM, (**E**) Bland–Altman plot of LVEDV by 2DTTE versus DHM, (**F**) Bland–Altman plot of LVESV by 2D TTE versus DHM, (**G**) Bland–Altman plot of LVEF by 2DTTE versus DHM, (**H**) Bland–Altman plot of LA volume by 2D TTE versus DHM.

**Figure 3 jcm-10-05030-f003:**
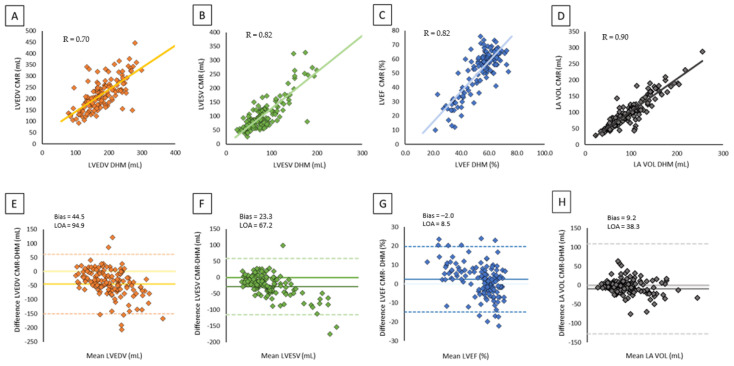
Correlation and agreement between CMR and DHM on left chamber volumes and LV function: (**A**) scatter plot of LVEDV by CMR versus DHM, (**B)** scatter plot of LVESV by CMR versus DHM, (**C**) scatter plot of LVEF by CMR versus DHM, (**D**) scatter plot of LA volume by CMR versus DHM, (**E**) Bland–Altman plot of LVEDV by CMR versus DHM, (**F**) Bland–Altman plot of LVESV by CMR versus DHM, (**G**) Bland–Altman plot of LVEF by CMR versus DHM, (**H**) Bland–Altman plot of LA volume by CMR versus DHM.

**Figure 4 jcm-10-05030-f004:**
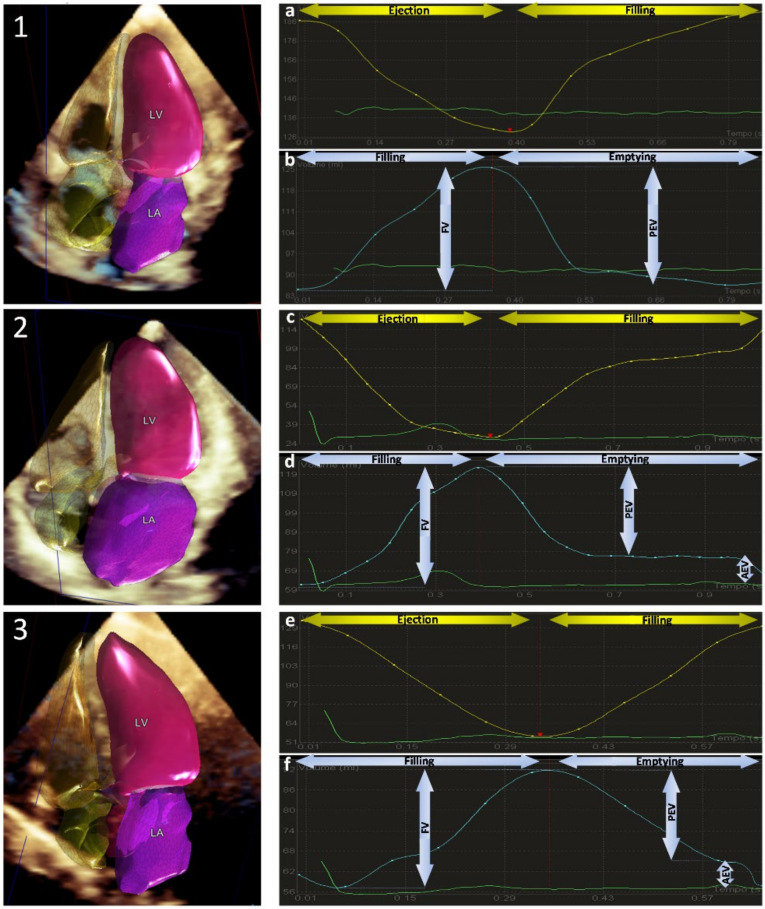
Analysis of left chamber volume–time curves in different pathological cases. Left panels: 3D chamber reconstructions. Right panels: LV (**top**) and LA (**bottom**) time–volume curves analysis. **1**. Atrial fibrillation: pathological absence of atrial contraction (atrial kick) is observed in both LV and LA time–volume panels (**a**,**b**). **2**. Severe mitral regurgitation: a consistent LA enlargement is clearly identified in the software reconstruction; LV maintains a preserved stroke volume with rapid ejection time (**c**); the enlarged atrium preserves its emptying properties, as sinus rhythm is guaranteed (**d**). **3**. Severe aortic stenosis: late systolic peak of LV curve is well represented (**e**); prolonged atrial passive emptying is observed as a result of marked increase in LV pressure (**f**). LA = left atrium; LV = left ventricle; FV = filling volume; PEV = passive emptying volume; AEV = active emptying volume.

**Figure 5 jcm-10-05030-f005:**
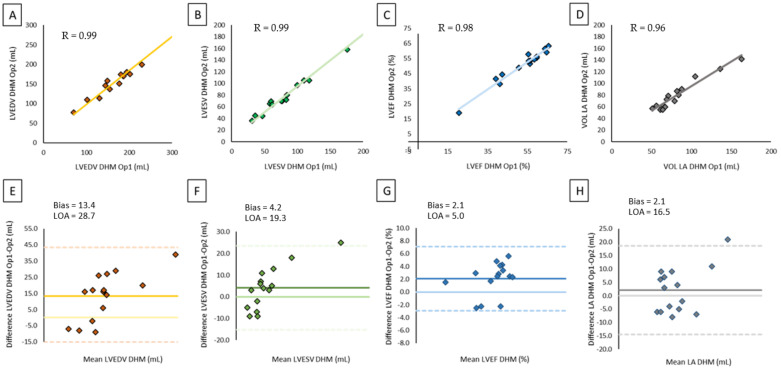
Intraobserver and interobserver variability.

**Table 1 jcm-10-05030-t001:** Study population characteristics. TTE: transthoracic echocardiography; CMR: cardiac magnetic resonance.

	2D TTE	CMR
Number of patients (*n*)	600	140
Age (y)	63 ± 14	59 ± 14
Female, *n* (%)	169 (28%)	34 (24%)
BSA (*n*)	1.87 ± 0.2	1.88 ± 0.20
Normal subjects, *n* (%)	40/600 (7%)	3/140 (2%)
Atrial fibrillation, *n* (%)	73/600 (12%)	6/140 (4%)
Valvular disease, *n* (%)	312/600 (52%)	80/140 (57%)
Coronary artery disease, *n* (%)	58/600 (10%)	6/140 (4%)
Dilated cardiomyopathy, *n* (%)	113/600 (19%)	35/140 (25%)
Miscellaneous, *n* (%)	25/600 (4%)	10/140 (7%)

**Table 2 jcm-10-05030-t002:** Feasibility of DHM, image quality, need for adjustments and time required for measurements in global population and in each subgroup. DHM: dynamic heart model. * *p* < 0.05 vs. normal subject.

		DHM Quality		Adjustment		Time Required (sec)
	Feasibility	Good	Suboptimal	Minor	Major	
Total of patients (*n*, %)	522/600 (87%)	327/522 (62%)	195/522 (28%)	149/522 (29%)	38/522 (6%)	41.3 ± 12.5
Normal subjects (*n*, %)	39/40 (97%)	23/39 (57%)	16/39 (40%)	9/39 (21%)	1/39 (3%)	31.2 ± 7.9
Atrial fibrillation (*n*, %)	59/73 (81%) *	28/59 (47%)	31/59 (53%)	15/59 (25%)	6/59 (10%)	45.7 ± 14.1
Valvular disease (*n*, %)	271/312 (87%)	120/271 (%)	151/271 (%)	65/271 (24%)	16/271 (6%)	44.5 ± 8.2
Coronary artery disease (*n*, %)	47/58 (81%) *	26/47 (46%)	21/47 (37%)	16/47 (34%)	5/47 (11%)	35.2 ± 10.2
Dilated cardiomyopathy (*n*, %)	98/113 (87%)	56/98 (57%)	42/98 (43%)	51/98 (52%)	12/98 (12%)	47.4 ± 16.3
Miscellaneous (*n*, %)	24/25 (96%)	18/24 (75%)	6/24 (25%)	5/24 (21%)	3/24 (12%)	38.5 ± 11.2

**Table 3 jcm-10-05030-t003:** Mean values of left chamber volumes and ejection fraction measured using automated software, conventional volumetric approach, and cardiac magnetic resonance. Correlations between DHM with the other techniques. TTE = transthoracic echocardiography; DHM = dynamic heart model; CMR = cardiac magnetic resonance; LOA = limits of agreement.

*N* = 600 patients	2D TTE	DHM	Correlation	Bias	LOA
LVEDV (mL)	141.9 ± 58.4	172.6 ± 63.8	0.94	30.7	47.6
LVESV (mL)	67.2 ± 49.8	82.8 ± 53.9	0.96	15.6	30.2
LVEF (%)	55.8 ± 13.7	54.6 ± 12.4	0.90	−1.2	9.2
LA volume (mL)	89.0 ± 38.4	90.7 ± 37.4	0.79	5.5	69.7
b. *N* = 140 patients	CMR	DHM	Correlation	Bias	LOA
LVEDV (mL)	226.1 ± 81.0	184.7 ± 55.8	0.70	−40.3	94.9
LVESV (mL)	115.2 ± 74.8	88.4 ± 43.3	0.82	−23.3	67.2
LVEF (%)	50.9 ± 15.4	53.3 ± 11.7	0.82	2.0	8.5
LA volume (mL)	101.3 ± 45.4	90.7 ± 37.4	0.90	−9.2	38.1

## Data Availability

Not applicable.

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
