# Peer review of "Feasibility and Accuracy of the Automated Software for Dynamic Quantification of Left Ventricular and Atrial Volumes and Function in a Large Unselected Population"

_jcm, 2021, doi:10.3390/jcm10215030_

Round 1
Reviewer 1 Report
This study aimed to evaluate the feasibility and accuracy of machine learning based automated dynamic quantification of left ventricular (LV) and left atrial (LA) volumes in an unselected population (12% in atrial fibrillation)
They conclude that the DHM software is feasible, accurate and quick in a large series of 26 unselected patients, including those with suboptimal 2D images or in atrial fibrillation.
The results are sound, and appropriate. The first study to test the feasibility and accuracy of this automated software in a large unselected population.
Ref 24 in wrong format
Author Response
Thank you for the comments.
We will change the Reference #24 in the text.
Reviewer 2 Report
Interesting and well designed study.
Author Response
Thank you
Reviewer 3 Report
This study evaluates the feasibility and accuracy of a machine learning based automated dynamic quantification of left ventricular (LV) and left atrial (LA) volumes in an unselected population.
The authors concluded that this Dynamic Heart Model (DHM) software is indeed accurate and quick in a large series of patients, including those with suboptimal 2D images or in atrial fibrillation.
In general, it was a novel study, evaluating a new imaging technique. Specifically, this was the first study to test this automated software in a large, unselected population. The authors found great correlations between 2D and DHM measures which shows promise to incorporate this technique in future medical practice. I suggest that this paper should be accepted for publishing after revising these following comments:
- The authors did not specify at what HR was the 2D echocardiography done, or whether they didn’t control for this. This could change cardiac volume and bias the ejection fraction value.
- Can the authors specify how many operators performed the echocardiography? And what was the operators’ level of experience?
- On what basis were DHM data divided into three main categories? Was it categorized by a single examinator? This qualitative and subjective qualification can cause major biases in the results.
- Were the manual epicardial and endocardial adjustments performed by one or multiple operators?
- Can the authors explain why and in which patients there were incorrect automatic identification of LV/LA borders?
- Page 4 line 188: the authors say that adjustments needed were higher in dilated cardiomyopathy vs normal subjects even though the difference is not significant. This statement should be revised.
- The authors specified that there were no significant differences between LA volume on 2DTTE and DHM even though the difference was close to 10%. Can the authors add p-values to support their statement?
- The authors mentioned that although there is a large difference in LVEDV between the different techniques, a correlation existed. However, how can a clinician determine the absolute exact value of LVEDV when managing a patient? Or should the physician specify every time the technique used?
- It would be interesting to compare DHM data between different pathology subgroups.
